# Current Perspectives on Endoscopic Nasobiliary Drainage: Optimizing Patient Management and Preventing Complications

**DOI:** 10.3390/jcm15010169

**Published:** 2025-12-25

**Authors:** Angelica Toppeta, Mattia Corradi, Beatrice Mantia, Adelaide Randazzo, Mario Schettino, Stefania De Lisi, Stefania Carmagnola, Raffaele Salerno

**Affiliations:** 1Diagnostic and Operative Endoscopy Unit, Azienda Socio Sanitaria Territoriale Fatebenefratelli Sacco Hospital, 20157 Milan, Italy; 2Department of Gastroenterology, Università degli Studi di Milano, 20122 Milan, Italy

**Keywords:** endoscopic nasobiliary drainage (ENBD), endoscopic retrograde cholangiopancreatography (ERCP), endoscopic biliary stenting (EBS), percutaneous transhepatic biliary drainage (PTBD), malignant biliary obstruction

## Abstract

Endoscopic nasobiliary drainage (ENBD) is a well-established technique for biliary decompression in both benign and malignant conditions. Over the past decades, its role has been extensively evaluated in comparison with endoscopic biliary stenting and percutaneous transhepatic biliary drainage. ENBD provides distinct clinical advantages, including real-time monitoring of bile output, the possibility to perform irrigation, and the ability to collect bile samples for cytological analysis. However, it also presents specific challenges such as patient discomfort, tube dislodgement, and the need for careful maintenance. This narrative review synthesizes current evidence from randomized controlled trials, retrospective cohorts, systematic reviews, and meta-analyses, highlighting the main indications, technical innovations, comparative outcomes with alternative drainage techniques, and strategies to prevent complications. Furthermore, it discusses emerging approaches aimed at improving patient tolerance, procedural efficiency, and environmental sustainability, offering an updated framework for optimizing patient management in both benign and malignant biliary obstruction.

## 1. Introduction

Biliary obstruction may arise from a wide range of etiologies, including malignant biliary obstruction (MBO), distal cholangiocarcinoma, pancreatic head tumors, bile duct stones, and benign strictures. Without timely intervention, patients are at risk of recurrent cholangitis, sepsis, hepatic dysfunction, and poor postoperative outcomes [1,2]. The management of biliary obstruction has become increasingly complex over time, reflecting advances in endoscopic technology, evolving surgical strategies, and growing emphasis on individualized patient care. Multiple drainage modalities, including endoscopic nasobiliary drainage (ENBD), endoscopic biliary stenting (EBS), and percutaneous transhepatic biliary drainage, are currently available, each with distinct technical characteristics, indications, and risk profiles. ENBD is an essential technique in the management of biliary obstruction. Its clinical utility extends across multiple scenarios from acute cholangitis and choledocholithiasis to MBO and postoperative bile leakage. ENBD provides an external drainage route that allows real-time monitoring of bile flow, repeated cholangiography, bile sampling for cytology, and continuous biliary irrigation [3]. Since its introduction by Cotton et al. in 1979 [4], the role of ENBD has evolved in terms of indications, design, and technical execution, positioning it as a key component in endoscopic biliary drainage strategies. Eastern centers, particularly in Japan and Korea, have traditionally favored ENBD, whereas Western institutions more frequently employ endoscopic biliary stenting (EBS) or percutaneous transhepatic biliary drainage (PTBD) [5,6,7,8,9].

## 2. Methods

This narrative review was conducted through a comprehensive and structured literature search aimed at summarizing current evidence on ENBD. A search of PubMed/MEDLINE, Embase, and the Cochrane Library was performed to identify relevant articles published up to June 2025. Search terms included combinations of “endoscopic nasobiliary drainage”, “ENBD”, “biliary drainage”, “endoscopic biliary stenting”, “percutaneous transhepatic biliary drainage”, “PTBD”, “ERCP”, “cholangitis”, and “malignant biliary obstruction”. Randomized controlled trials, observational studies, systematic reviews, meta-analyses, and relevant clinical guidelines were considered. Additional articles were identified through manual screening of reference lists of key publications. Studies were selected based on relevance to clinical indications, technical aspects, comparative outcomes, and complications of ENBD in both benign and malignant biliary diseases. Non-English articles, case reports with very small sample sizes, and studies lacking sufficient methodological detail were excluded. Data were extracted qualitatively and synthesized narratively, with particular attention to guideline recommendations, comparative effectiveness with alternative drainage techniques, and recent technical innovations. Our aim is to provide clinicians with an up-to-date and comprehensive synthesis of current knowledge on ENBD, integrating data from both Eastern and Western literature. By analyzing clinical evidence, technical developments, and comparative outcomes, it seeks to guide evidence-based decision-making to optimize patient management and long-term outcomes.

## 3. Indications and Clinical Applications

### 3.1. Choledocholithiasis and Acute Cholangitis

Choledocholithiasis represents one of the most common indications for endoscopic retrograde cholangiopancreatography (ERCP). Standard ERCP techniques include endoscopic sphincterotomy (EST), endoscopic papillary balloon dilation (EPBD), stone extraction using a balloon or basket, and biliary drainage using either an ENBD or stent placement [2,3,5,10]. However, therapeutic maneuvers such as EST, EPBD, and stone extraction are associated with procedure-related risks, including post-ERCP pancreatitis and biliary sepsis [5,6,10]. In cases complicated by infection, ENBD plays a particularly valuable role (Table 1). The Tokyo Guidelines 2018 and the American Society for Gastrointestinal Endoscopy (ASGE) guidelines recommend prompt biliary drainage in patients with acute cholangitis, favoring ENBD or EBS over percutaneous approaches [5,6]. The ESGE guidelines similarly recognize ENBD as a valid drainage option in selected patients [10]. Several studies have demonstrated that ENBD can reduce the incidence of post-ERCP complications, particularly in patients with retained stones, infected bile, or biliary blood clots. By lowering intraductal pressure and allowing for bile decompression and continuous irrigation, ENBD reduces bacterial and endotoxin translocation, thereby improving clinical outcomes [1,5,11,12,13,14,15,16]. Moreover, nasobiliary cholangiography is often employed to confirm complete stone clearance, especially in cases with severe pneumobilia, ductal dilation, or following lithotripsy [12]. Xu et al. demonstrated that ENBD placement following papillary balloon dilation significantly reduces the incidence of post-ERCP pancreatitis [13]. A recent systematic review and meta-analysis by Yin et al. provided evidence that the omission of ENBD after elective endoscopic clearance of choledocholithiasis is not associated with increased postoperative complications or adverse outcomes [17]. However, the analysis excluded patients with severe emergencies such as acute suppurative cholangitis, acute pancreatitis, or severe symptomatic jaundice, for whom biliary drainage remains warranted according to current international guidelines [5,6,10]. Although multiple studies report potential benefits of ENBD in infection control, biliary decompression, and post-ERCP management, the overall evidence remains heterogeneous, with significant variability in study design, patient selection, and clinical context. These factors should be considered when interpreting outcomes.

### 3.2. Malignant Biliary Obstruction and Liver Transplantation

Malignant biliary obstruction may arise at intrahepatic, hilar, or extrahepatic levels; intrahepatic forms can develop from congenital ductal anomalies or long-standing lithiasis [18]. ENBD plays an important role in malignant hilar obstruction (MHO), particularly in candidates for surgical resection. Preoperative drainage reduces cholestasis and cholangitis risk and can support functional optimization of the future remnant liver. Beyond decompression, ENBD also offers diagnostic value: Yagioka et al. showed that bile cytology via ENBD increases the sensitivity for malignancy [19]. In a cohort of 164 patients with suspected perihilar cholangiocarcinoma, Kawashima et al. demonstrated that unilateral drainage of the future remnant lobe is effective even in Bismuth–Corlette type III–IV strictures, and that avoiding sphincterotomy and pancreatography may help reduce procedure-related complications [20]. Several studies have reported lower infectious complications with ENBD compared with PTBD, which also carries a risk of tumor seeding [7,8].

ENBD additionally facilitates surgical planning. ENBD-CT cholangiography enables precise mapping of biliary anatomy, including caudate lobe and complex variants, as shown by Akita et al., while Kiritani et al. demonstrated that placing the catheter near the hilum improves delineation of proximal ducts [21,22]. Reflecting this evidence, the Japanese Society of Hepato-Biliary-Pancreatic Surgery recommends ENBD for selective drainage of the future remnant lobe, often in combination with volumetry or portal vein embolization [5]. ASGE guidelines likewise include ENBD as an acceptable endoscopic option for preoperative biliary drainage in MHO [23]. Meta-analyses suggest that ENBD may reduce infectious complications and postoperative bile leaks compared with EBS [9,18], although recent multicenter propensity-matched data indicate similar efficacy and safety between ENBD and EBS, with the preferred technique depending on patient factors and institutional expertise [24].

After liver transplantation, ENBD remains particularly useful for managing bile leaks by allowing rapid external control and continuous monitoring. Studies comparing ENBD and EBS for post-transplant leaks show similar clinical success, although sample sizes are small and follow-up is limited [25,26,27,28]. Overall, ENBD is a safe and effective option in the preoperative and postoperative management of MHO, but most comparative studies are retrospective and primarily from East Asian centers, where surgical timelines differ substantially from Western practice. These factors limit generalizability and underscore the need for prospective multicenter studies to clarify which patients benefit most from ENBD.

### 3.3. ENBD in Other Clinical Contexts

Although less common, ENBD has demonstrated utility in several uncommon or complex scenarios. It has also been used for the management of postoperative bile leaks after cholecystectomy, particularly in cases where stent placement is not feasible or when continuous monitoring of bile output is needed [29]. Ikeda et al. reported successful use of combined ENBD and nasopancreatic drainage in refractory duodenal ulcers, facilitating decompression and continuous drainage of bile and pancreatic secretions, thereby promoting mucosal healing in cases unresponsive to standard medical therapy [30]. Ahmed et al. described ENBD as an effective option for treating intractable pruritus in patients with cholestatic liver disease, especially in those with hereditary cholestatic disorders or drug-induced cholestasis, by providing rapid symptom relief through temporary external biliary diversion and reduction in circulating pruritogens. ENBD has also been used during pregnancy where medical management has failed [31]. According to the Tokyo Guidelines 2018, ENBD may also be feasible in patients with altered surgical anatomy, provided that balloon-assisted enteroscopy ERCP is possible [5]. Finally, Lou et al. reported a novel combined gastroscopic and choledochoscopic technique for ENBD performed during laparoscopic common bile duct exploration. This method allowed direct endoscopic visualization for catheter placement and ensured secure postoperative drainage, effectively preventing bile leakage and eliminating the need for T-tube insertion [32].

## 4. Technical Aspects and Innovations

ENBD is usually performed using 5–7 Fr catheters placed transnasally under fluoroscopic guidance during ERCP. The procedure involves advancing a drainage tube into the common bile duct and externalizing it through the nasal cavity after passing the duodenum, stomach, esophagus, pharynx, and throat [4]. The critical step is to advance the catheter tip under fluoroscopy to an optimal position within the biliary duct to ensure effective bile drainage. Following successful cannulation, the tube is converted from an orobiliary to a nasobiliary route and securely fixed to the nasal ala and/or cheek.

Traditionally, this conversion was achieved by inserting a 16F clear polyethylene carrier tube (a shortened soft nasogastric or nasopharyngeal tube) through the nose until its distal end reached the posterior pharynx The endoscopist then grasped the tube through the mouth, pulled it out orally, attached the proximal end of the biliary tube to the carrier, and withdrew both through the nasal passage, completing the conversion into a nasobiliary tube. Although effective, this technique was technically demanding and uncomfortable for patients. To simplify this step, Li et al. introduced a novel adjustable snare technique in a randomized controlled trial of 210 patients, achieving higher first-pass success rates, shorter procedure times, and greater patient comfort compared with conventional guidewire-assisted methods [33].

Shah and Barkin introduced a “hands-off” technique for converting an orobiliary to a nasobiliary drainage tube without endoscopic reinsertion, improving procedural safety and patient tolerance [34]. Similarly, Watanabe et al. proposed a magnet-assisted method enabling magnetic coupling of oral and nasal catheters for smooth and atraumatic tube externalization [35].

Mi et al. demonstrated that modified fixation techniques significantly reduce both unintentional removal and tube occlusion [36]. In addition, new hybrid systems such as the UMIDAS NB, which combines an ENBD catheter with a plastic stent, allowing for internal drainage after external tube removal [37]. Clinical studies have shown that this system is particularly useful in cases of cholangitis where initial external monitoring is required but long-term internal drainage is desired. A Korean pilot study confirmed that UMIDAS stents can reduce unplanned reinterventions by maintaining biliary decompression after tube removal [38].

In a randomized trial, Fujisawa et al. found that 7 Fr catheters provide faster relief of obstructive jaundice, whereas 5 Fr catheters offer comparable efficacy with better patient tolerance in non-urgent situations [39]. Gong et al. identified both the type of nasobiliary tube and the extracorporeal length as independent risk factors for tube migration, noting that maintaining an external length greater than 150 cm can significantly reduce the risk of dislodgement [40].

Furthermore, nasobiliary cholangiography via ENBD is often used to verify complete stone clearance; however, its accuracy may be limited in cases with small residual stones (<4 mm) or marked ductal dilation [41]. The procedure also exposes patients and staff to radiation and carries potential risks of allergic or toxic reactions to iodinated contrast agents. To mitigate these limitations, Wang et al. proposed the use of contrast-enhanced ultrasound (CEUS)-guided contrast injection via the ENBD tube. This approach allows for radiation-free, cost-effective, and dynamic visualization of residual stones after ERCP [12]. Similarly, Wu et al. demonstrated that ENBD-based saline-injection ultrasound is an inexpensive, repeatable, and noninvasive technique for detecting remnant common bile duct stones, improving postoperative follow-up accuracy [42].

## 5. Comparative Evidence: ENBD vs. EBS vs. PTBD

Choosing the optimal drainage modality requires balancing efficacy, safety, patient comfort, and procedural feasibility. ENBD, EBS, and PTBD each have distinct advantages and limitations depending on the clinical setting and patient condition.

### 5.1. ENBD vs. Endoscopic Biliary Stenting (EBS)

Several meta-analyses and multicenter studies have demonstrated that ENBD may lower the incidence of infectious complications compared with EBS [9,24]. Early evidence from Japan, particularly from Kawakami et al., favored ENBD in perihilar disease, while more recent propensity-matched analyses reported comparable efficacy and safety between the two methods [8,24]. ENBD enables external control and repeated cholangiography, which are critical advantages in the preoperative period or in patients with infected bile. Conversely, EBS is associated with higher rates of stent occlusion due to biofilm formation, resulting in cholangitis or recurrent obstruction [24,43].

Fujii et al. demonstrated that bile cultures were more frequently positive in patients undergoing EBS than ENBD, correlating with a higher rate of postoperative abdominal abscesses after pancreatoduodenectomy [43]. Similarly, Zhang et al. observed comparable overall postoperative outcomes between the two techniques but noted a higher rate of deep abdominal infections in the EBS group, with no difference in wound or pulmonary infections [44]. Although ENBD generally shows lower morbidity compared with PTBD, the supporting evidence is similarly constrained by observational designs and non-standardized indications. These limitations prevent definitive conclusions and highlight the importance of context-specific decision-making. Given the predominantly observational nature of the evidence, neither modality has demonstrated clear superiority across all scenarios. In clinical practice, ENBD is often preferred when short-term monitoring is required, whereas EBS remains advantageous for long-term internal drainage and better patient tolerance.

### 5.2. ENBD vs. Percutaneous Transhepatic Biliary Drainage (PTBD)

PTBD achieves decompression through percutaneous catheterization of the intrahepatic ducts under ultrasound or fluoroscopic guidance. It can be performed without general anesthesia and is often available even in non-tertiary settings. However, PTBD carries several limitations, including bleeding, catheter tract infection, discomfort, and the risk of tumor seeding, particularly in malignant biliary obstruction.

Comparative studies and meta-analyses indicate that ENBD offers similar technical success and drainage efficacy to PTBD but may be associated with fewer infectious and hemorrhagic complications, shorter hospitalization, and better quality of life [8,9,45,46]. Similarly, Zhou et al. reported that in patients with advanced hilar cholangiocarcinoma, ENBD achieved comparable biliary decompression with fewer hemorrhagic and septic complications, shorter recovery time, and no risk of tumor seeding [24]. Nevertheless, the available evidence is predominantly retrospective, and outcomes vary according to institutional expertise, tumor anatomy, and drainage strategy.

However, PTBD remains indispensable in certain scenarios, particularly in cases with altered anatomy, high-level or complex strictures, or failed endoscopic access, where endoscopic approaches are technically unfeasible. The choice between ENBD and PTBD should therefore be individualized, taking into account anatomy, institutional expertise, and patient condition.

## 6. Complications and Risk Mitigation

Although ENBD is generally safe and effective, several complications and limitations must be considered to optimize patient care. Because the drainage tube passes through the nasal cavity, ENBD frequently causes discomfort such as nasal irritation, sore throat, or sleep disturbances and requires continuous external maintenance to ensure patency. These factors can reduce patient compliance, particularly during prolonged drainage, and make ENBD unsuitable for patients with abnormal nasal anatomy, coagulopathy, frailty, or impaired consciousness [33,39]. Tube removal may occasionally cause excessive secretions or pharyngeal bleeding during tube removal, which can increase the risk of pulmonary infection if aspirated, particularly in frail or sedated individuals [17]. Long-term indwelling may result electrolyte disturbances, or mechanical complications due to hepatointestinal motion. Moreover, ENBD placement increases radiation exposure time for both patients and medical staff [14]. Infectious complications may result from tube occlusion, contamination, or inadequate bile flow; structured nursing protocols—including routine flushing, bile output assessment, and secure fixation—can mitigate these risks but increase caregiver workload [17]. However, in cooperative and fully conscious patients, tolerance is markedly better when they are properly informed and when the tube is required only for a short duration.

## 7. Economic and Environmental Considerations

Economic and environmental factors are increasingly relevant in selecting the optimal biliary drainage modality. From a cost perspective, plastic stents are inexpensive and easy to deploy but require frequent replacement, leading to higher cumulative costs in long-term management. Self-expandable metal stents (SEMS) have higher upfront costs but superior patency, fewer reinterventions, and overall better cost-effectiveness in patients with longer survival [45]. ENBD devices are relatively low-cost. In contrast, percutaneous transhepatic biliary drainage (PTBD) generally entails the highest total cost owing to procedural complexity, imaging requirements, and external catheter care.

Beyond financial considerations, environmental sustainability has become integral to modern healthcare economics. The concept of green endoscopy emphasizes reduction in medical waste, energy consumption, and the carbon footprint of endoscopic practice. Compared with PTBD, which involves multiple radiologic procedures and disposable external catheters, endoscopic approaches such as ENBD and internal stenting may offer a lower environmental impact when successful in a single session. Nonetheless, the reliance on single-use plastic components remains a major limitation. Future developments in biodegradable stents, recyclable materials, and streamlined workflow design are essential to align biliary drainage practices with both cost efficiency and environmental responsibility [47,48].

## 8. Discussion

This narrative review provides an integrated overview of the current evidence on endoscopic nasobiliary drainage and places its use within the broader evolution of biliary drainage strategies [1,4]. Rather than supporting a universal preference for ENBD, the available data suggest that its clinical value is highly context-dependent and closely linked to specific procedural goals, such as short-term decompression, infection control, and the need for repeated biliary assessment [6,10].

A critical appraisal of the literature reveals substantial heterogeneity in study design, patient populations, and clinical pathways, which limits direct comparison between ENBD, endoscopic biliary stenting, and percutaneous transhepatic biliary drainage [9,24,25,26]. Differences in disease severity, timing of intervention, preoperative waiting periods, and peri-procedural management likely contribute more to outcome variability than the drainage modality itself [20,23,24]. Consequently, reported advantages of ENBD in reducing infectious complications should be interpreted with caution, particularly when extrapolating results across different healthcare settings.

From a practical standpoint, ENBD appears most advantageous in clinical scenarios where active management of the biliary system is required, including ongoing irrigation, monitoring of bile characteristics, and confirmation of ductal clearance. In contrast, its role is more limited in settings requiring prolonged drainage, where patient comfort, risk of accidental dislodgement, and resource utilization become increasingly relevant considerations [40,41]. These trade-offs underscore the importance of aligning the drainage strategy with the anticipated duration and objectives of treatment [44,49].

Overall, the evidence supports a selective and indication-driven use of ENBD within modern biliary drainage algorithms, favoring individualized decision-making over routine application.

## 9. Future Perspectives

Future developments in ENBD are expected to focus on enhancing patient comfort, procedural efficiency, and clinical precision. Advances in catheter materials, fixation systems, and hybrid drainage devices may improve tolerability and reduce complications such as tube migration and occlusion. Biodegradable and self-flushing catheters could further simplify postoperative management, particularly in short-term drainage settings.

Hybrid drainage systems, such as stent–catheter combinations, represent a promising evolution that integrates the advantages of internal and external drainage. These systems allow close monitoring during the acute phase while maintaining long-term decompression after tube removal.

Concurrently, artificial intelligence (AI)–driven decision support models are emerging as valuable tools for personalized management. By integrating clinical, imaging, and microbiological data, AI algorithms may help predict which patients are most likely to benefit from ENBD, EBS, or PTBD, thereby improving outcomes and minimizing unnecessary interventions.

Large-scale, multicenter randomized trials and long-term outcome studies remain essential to establish standardized protocols for ENBD use across diverse clinical contexts. The ongoing refinement of endoscopic technology, combined with precision medicine approaches, will continue to redefine the role of ENBD in contemporary hepatobiliary care.

The management of bile leaks after liver transplantation illustrates this evolving paradigm. ERCP with stent placement or ENBD has become a key minimally invasive approach, reducing surgical intervention and patient morbidity. Although modern endoscopic techniques have achieved outcomes comparable to surgery, challenges persist in complex cases such as large anastomotic leaks or altered anatomy. Robust multicenter trials, technological innovation, and precision-medicine frameworks are needed to overcome these limitations.

## 10. Conclusions

ENBD remains a versatile and valuable technique for the management of biliary obstruction. Its capacity to allow real-time monitoring, bile irrigation, and cytologic sampling offers unique advantages in both benign and malignant conditions. Ongoing innovations in device design, fixation systems, and hybrid drainage technologies are progressively mitigating its limitations, enhancing safety and procedural efficacy. With continued refinement and evidence-based integration into multidisciplinary care, ENBD is poised to retain a pivotal and evolving role in modern hepatobiliary practice.

## Figures and Tables

**Table 1 jcm-15-00169-t001:** Main Indications and Clinical Applications of Endoscopic Nasobiliary Drainage.

Clinical Setting	Main Indications/Scenarios
Choledocholithiasis and Acute Cholangitis	Biliary drainage after ERCP for choledocholithiasis, especially with infection or incomplete clearance. Management of acute cholangitis according to Tokyo Guidelines 2018, ASGE, and ESGE recommendations.
Malignant Biliary Obstruction (MBO)	Preoperative drainage in perihilar or distal cholangiocarcinoma. Bridge to surgery or liver transplantation. Alternative to PTBD or EBS in infection-prone patients.
Liver Transplantation	Treatment of bile leaks and anastomotic strictures after liver transplantation. Postoperative monitoring of bile flow.
Postoperative Bile Leaks (Non-transplant)	Post-cholecystectomy or biliary surgery bile leaks, especially when stent placement is not feasible.
Refractory Duodenal Ulcer	Adjunctive therapy in refractory or penetrating duodenal ulcer associated with bile reflux.
Cholestatic Liver Disease/Intractable Pruritus	Temporary external biliary diversion to relieve pruritus in intrahepatic or drug-induced cholestasis.
Pregnancy (Selected Cases)	Biliary obstruction unresponsive to medical therapy.
Altered Surgical Anatomy	Biliary drainage in patients with Roux-en-Y or other reconstructions (balloon-assisted enteroscopy ERCP).

## Data Availability

Not applicable.

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
