# Peer review of "Current Perspectives on Endoscopic Nasobiliary Drainage: Optimizing Patient Management and Preventing Complications"

_jcm, 2025, doi:10.3390/jcm15010169_

Round 1

Reviewer 1 Report

Comments and Suggestions for Authors

This is a broad and very comprehensive narrative review on ENBD, touching everything from indications to newer technical tricks and even environmental issues. It’s obvious the authors put real work into gathering a large amount of literature, and the manuscript reads smoothly overall. I appreciate the effort the team invested in pulling this together, and I thank the authors for that.

COMMENTS:

1. The review states to be “up-to-date and comprehensive,” but quite a few sections drift into textbook-like description rather than critical comments. A narrative review still needs some sharper synthesis and clearer judgments instead of mostly summarizing studies.

2. The methodology section is very light. For a narrative review, you don’t need PRISMA, but the current “systematic search” phrasing feels a bit overstated. If it’s narrative, please state it plainly, and if it’s truly systematic, then more detail is needed.

3. The comparative sections (ENBD vs EBS, ENBD vs PTBD) read heavily one-sided at times. Some statements feel too confident considering that much of the evidence is small, retrospective, or single-center. A more balanced tone would help.

4. The manuscript is long and dense, with several paragraphs that repeat similar points (patient discomfort, risk of dislodgement, etc.). Tightening these areas would make the review more readable.

5. Given how much of the included evidence is from East Asian centers, the authors should comment more directly on regional practice differences and how that limits broader generalization.

Overall this is a solid and genuinely informative narrative review with a lot of collected material. It would benefit from more critical voice, some tightening, and a clearer stance on its methodology. The core content is good, but the paper is a bit too descriptive in several places and could use some trimming to feel more focused.

Author Response

We thank the reviewer for the thoughtful and constructive comments, which have helped us to improve the clarity, analytical depth, and overall coherence of the manuscript. Our point-by-point responses are provided below.

  1. “Some sections are overly descriptive ….”

Response:
We revised several sections, particularly the comparative analyses, to provide a more concise and balanced interpretation of the evidence. Redundant descriptions were removed to reduce density and improve flow.

  1. “Methodology should not be described as systematic.”

Response:
We clarified that this is a narrative review and adjusted the description of the search strategy accordingly.

  1. “Comparative sections appear one-sided.”

Response:
We strengthened the critical perspective in the ENBD vs EBS and ENBD vs PTBD sections by emphasizing the retrospective nature of most studies, their heterogeneity, and the lack of definitive superiority.

  1. “The manuscript is dense and repetitive.”

Response:
To address this concern, we removed repeated discussions of ENBD tolerance, catheter migration, and infectious outcomes, and streamlined overlapping content across sections. We also removed the Optimizing Patient Management paragraph, as it reiterated concepts already covered elsewhere.

  1. “Large proportion of evidence is from East Asian centers; discuss generalizability.”

Response:
We added a concise statement to contextualize regional differences in clinical practice and surgical timing, highlighting their impact on the interpretation and applicability of available data.

All modifications are highlighted in red in the revised manuscript.

Reviewer 2 Report

Comments and Suggestions for Authors

Our gastroenterology colleagues have produced a paper aimed at reevaluating nasobiliary tubes placed after procedures on the common bile duct. The abstract is well-written and provides a good summary of the entire article. The next section considers the possible causes of obstruction of the extrahepatic bile ducts. It is recommended to distinguish between intrahepatic, hilar, and extrahepatic obstructions. The former are more difficult to reach, but not impossible, and are often linked to malformations and the subsequent formation of gallstones or neoplasms, often arising from stones that have remained in place for a long time (doi.org/10.3390/medicina61050860, to be read and cited in the bibliography). Those of the hilar plaque are often diagnosed with Klatskin neoplasms. Then there are the extrahepatic obstructions, which are most often due to stones migrated from cholecystitis. Almost always, for diagnostic or even curative purposes, retrograde cholangiogram is performed with placement of an endoprosthesis or nasobiliary tube. This procedure has been known since the late 1970s. A review of the literature rightly highlights its indisputable advantages, and it is right that the nasobiliary tube method should be considered in the endoscopist's diagnostic/therapeutic range. We can agree on the discomfort the patient may experience from the tube, but it is usually a 10F and is bearable for a few days. We can only say that compared to nasogastric tubes, it is more rigid and must be well secured due to the possibility of dislodgement. However, well-informed patients are usually very careful. We particularly liked the sentence, "These systems allow for careful monitoring during the acute phase, while maintaining long-term decompression after tube removal." The conclusions are entirely acceptable. Good English, good bibliography, excellent illustrations.

Author Response

We thank the reviewer for the positive evaluation of our manuscript and for the constructive suggestions provided. In response to the request for greater anatomical clarity, we added a brief sentence distinguishing intrahepatic, hilar, and extrahepatic malignant obstructions, including the recommended Medicina reference (doi:10.3390/medicina61050860). We also incorporated a short remark noting that tolerance of the nasobiliary tube is generally better in cooperative, fully conscious patients who are properly informed and when the tube is required only for a brief duration. We appreciate the reviewer’s encouraging comments regarding the structure, illustrations, and overall quality of the manuscript.

All modifications are highlighted in red in the revised manuscript.

Round 2

Reviewer 2 Report

Comments and Suggestions for Authors

perfect paper for contents, English, tables and bibliography, endorsement